# A theoretical framework for polarization as the gradual fragmentation of a divided society
Ana-Maria Bliuc [1] ✉, John M. Betts [2], Matteo Vergani[3], Ayoub Bouguettaya[4] & Mioara Cristea[5]

We propose a framework integrating insights from computational social science, political, and social psychology to explain how extreme polarization can occur in deeply divided societies. Extreme polarization in a society emerges through a dynamic and complex process where societal, group, and individual factors interact. Dissent at different levels of analysis represents the driver of this process, where societal-level ideological dissent divides society into opposing camps, each with contrasting collective narratives. Within these opposing camps, further dissent leads to the formation of splinter factions and radical cells—sub-groups with increasingly extreme views. At the group level, collective narratives underpinning group identity become more extreme as society fragments. At the individual level, this process involves the internalization of an extreme group narrative and norms sanctioning radical behavior. The intense bonding within these groups and the convergence of personal and group identities through identity fusion increase the likelihood of radical group behavior.

The rise of social media, increased online interaction, and greater access to information, as well as misinformation, have all contributed to societies becoming more polarized[1,2]. Problematic online encounters between both like-minded people and those with opposing views have been shown to result in the reinforcement of extreme views and intensification of belief polarization, reducing the willingness of groups to engage in constructive dialog[3-7]. In extreme cases, this can lead to radicalization, where individuals or groups adopt extreme views that sanction political violence[8-10]. An illustration of this process is the U.S. Capitol attack on January 6, 2021, which can be traced back to the formation of "Stop the Steal" Facebook group, in the aftermath of the 2020 U.S. Presidential Election. Quickly attracting over 300,000 members, the group amplified unfounded election fraud claims[10,11]. Within this group, dissenting views were excluded and member polarization increased[8,12,13], creating ideal conditions for the emergence of a divisive narrative, further inflamed by misinformation and conspiracy beliefs[14,15], and ultimately leading to radicalization and political violence[14]. This example also highlights the very real danger of polarization for democratic societies, and the importance of understanding the socio-psychological processes that underpin it, so that more effective evidence-based strategies to counter polarization are developed.

Despite the need for empirically grounded approaches to counteract polarization, the potential impact of existing research in this domain is reduced by insufficient crossover between the academic disciplines investigating this phenomenon. To address this limitation, we conducted an extensive review of research from social and political psychology, as well as the disciplinary domains under the umbrella of computational social science. By bringing together perspectives from these diverse disciplines to the study of polarization, we develop a conceptual framework of polarization as a layered, complex process that can ultimately lead to radicalization. To identify the mechanisms underpinning polarization from moderate levels to its most extreme forms, we draw on social psychological theories of group dynamics. We propose that the study of these processes is enriched by integrating approaches from computational social science, which enable a more refined empirical analysis of the (micro-level) interactions between individuals that underpin these dynamics[16,17], as well as the macro-level social structures that result[13,16,18]. The advances in our current understanding of polarization resulting from this integration would help establish a more robust foundation for policy recommendations aimed at mitigating the causes of societal polarization and potential radicalization of individuals[9,14].

At the core of our framework is the dynamic interplay between individual and collective processes within a divided society, when there is dissent over a contentious issue. Dissent is defined here as disagreement on a position on an issue which is sufficiently important to provide a basis for social identity formation[19,20]. Polarization is seen as a complex process driven by incremental changes in attitudes and beliefs in the individual, that are amplified through social influence and emergent group dynamics[15,16].

[1]Division of Psychology, University of Dundee, Dundee, UK. [2]Department of Data Science & AI, Monash University, Melbourne, VIC, Australia. [3]School of Humanities & Social Science, Deakin University, Melbourne, VIC, Australia. [4]Cedars-Sinai Medical Center, Los Angeles, US. [5]Department of Psychology, Heriot-Watt University, Edinburgh, UK. ✉e-mail: abliuc001@dundee.ac.uk

We propose that this process is not only initiated by dissent in a society, but it is also sustained by it. In other words, dissent is both the catalyst of polarization, as an initial difference in position possibly leading to the formation of oppositional identities, and its driver, sustaining and accelerating polarization in a society over time (as difference in position leads to further differentiation in all aspects of identity content, including beliefs, values, and norms, etc.).

We adopt a conceptualization of polarization as a process of increasing the ideological and psychological distance between groups underpinned by specific social identity content[19,21–23]. This conceptualization integrates aspects of both issue-driven polarization (where the distancing is driven by ideological dissent that may cross partisan boundaries) and affective polarization (where distancing is driven by animosity and hostility between opposing sides). These interconnected ideological and affective aspects manifest as forces pulling groups apart. Differences in ideological position become the driver of the process of sorting the public into opposing sides along the lines of conflicting multiple social identities; a process further enhanced by difference, distrust, and disdain for the opposing group[24,25]. Taking this view, polarization arises as a gradual extremization of the ingroup position and increased distancing from ideological outgroups. As the ingroup position becomes more extreme, the group becomes more ideologically segregated.

Group identification leads to affective polarization and distancing between the ingroup and outgroup/s in terms of group identity content. This, in turn, results in ideological shifts indicated by increased extremity of belief, distancing of position relative to others, and psychological changes such as increased hostility and mistrust towards outgroup/s[19,26–28]. The changes that fuel polarization are reflected in the evolution and refinement of a group's identity, and collective narratives[15,16]. An important point here is that we see polarization as having a dual pathway, being a process driven by both intergroup and intragroup interaction[29]. Fundamental to understanding this process is the recognition that polarization is an ongoing and transformative process in which collective as well as individual identities are in continuous evolution over time[30].

Polarization has been generally regarded as a continuous process that, once started, and left unchecked, progresses towards increased distancing between opposing sides and ultimately full segregation[31–34]. That is, the repeated process of ideological and affective distancing leads societies to fragment into groups having more extreme positions. While in reality, extreme polarization is relatively rare, conceptually, this process could continue until a limiting state of complete segregation between the polarized groups is achieved, and radical cells may form. To understand how to slow down or stop this process, we need to approach polarization as a complex and progressive process driven by many interconnected factors, transforming both the society and individuals. We identify three key mechanisms underpinning polarization, leading to distinct transformations and outcomes in the society or group concerned, each representing a more fragmented version of the preceding state, broadly speaking. These mechanisms are: (a) division into opposing ideological camps; (b) fragmentation within camps into dissenting factions; and (c) extreme clustering within factions. These processes result from the varying dynamics of individual and group-based transformations that drive individuals towards increasingly ideologically and psychologically distant positions. This understanding of polarization as a complex and gradual transformative process is in line with perspectives from political science that focus on radicalization as a pathway or a trajectory that may vary at different points in time and can be reversed via de-radicalization and disengagement[35,36]. These processes can be viewed as sequential, in that groups need to divide before they fragment and re-cluster, but not time-phased, in the sense that at a single point in time, variations of these processes can be happening simultaneously in different subgroups, within varying contexts.

## Society as a complex system

From a computational social science view, societies can be conceptualized as a complex system composed of dynamic elements in continuous interaction[37–39]. This is in line with structuration theory in sociology, which emphasizes the dynamic and continuous mutual influence between units of analysis (or agents, as an abstraction of individuals) in a society[40]. Consistent with this perspective, we propose that polarization can be understood as the result of an iterative process of social interaction resulting in beliefs converging and diverging, and the making and breaking of social connections that culminate in the clustering of extreme beliefs within a society. This outcome can be explained in terms of social emergence, a construct that refers to a collective phenomenon (polarization in this case) resulting from the actions of many, but which cannot be attributed to the action of any individual[37,41–43]. This conceptualization enables the application of computational perspectives where the process of polarization can be modeled in artificial societies by simulating interactions between autonomous entities (agents) with unique characteristics and predetermined rules. For instance, polarization can be studied by analyzing agent-based models, in which society is represented as a population of agents (holding a belief and/or intention), where simulated social interaction takes place, and the resulting relationship between agents can be regarded as forming a social network[44,45]. Through successive interactions between agents, these models can reproduce the emergence of complex outcomes within a system, such as opinion dynamics leading to extreme belief clustering within a society[46–48]. Because they enable the complex outcomes of repeated but explainable interactions to be observed, these models are very effective in connecting micro-level assumptions regarding individual agent behaviors to macro-level patterns in societies[10,49]. For instance, simulation models based on dynamic social impact theory reveal the influence of mutual social interactions on polarization and clustering[50,51]. More recently, opinion extremization has been modeled as exposure to mixed evidence on a complex issue where the initial opinion is endorsed through "biased assimilation", resulting in the opinion becoming more extreme[46].

This type of modeling offers another benefit, by allowing the role played by the network structure in dynamic processes (such as polarization) to be identified, by directly integrating different types of network structure in the study design. For example, systematic studies of complex dynamic societal change can be conducted, by comparing networks with weak ties to those with strong ties[52]. These models have proven useful in illustrating, for instance, the influence of cultural complexity involving a broader diversity of extreme views in countering opinion extremization[48,53]. In society, social identity content (such as beliefs and group norms) manifests in the way group members process relevant information, refine their beliefs, and interact with others, thus directly shaping the process of polarization. Recent models have incorporated these social identity dynamics to identify the role of social media and online interaction in increasing affective polarization, for example[25], or to determine the role of social norm compliance in the emergence of extreme ideas[54].

Although agent-based modeling is useful for understanding the complexities of opinion dynamics, many of these models fail to reflect the psychological factors that enable opinions to become such powerful drivers of behavior. In modeling polarization, including clustering and extremization, we argue that opinion is often treated as an entity that is disconnected from the complex social and personal identity of the agent. Although, this is a necessary simplification for the sake of model feasibility, a better integration of the connections between the development of extreme opinions and the individuals' propensity to engage in extreme actions within these models is needed. To increase the predictive power of agent-based models and their capacity to capture the psychological link between belief and action in the "real world", social psychological theory can be used to provide more robust and nuanced theoretical foundations for these models. Building on emerging work in this area[17,24,25,55,56], this approach would enable researchers to deduce the key characteristics and behavior of agents as group members to test specific propositions relating to intergroup and intragroup behavior, and social identity[57,58].

Unlike the computational social science approach, mainstream social psychology tends to view society as a superordinate social identity comprised of various groups and social categories with subordinate

identities[59–62]. Individuals are inherently situated within society but acquire their social identities through meaningful social categories and group memberships[63]. Recent contributions to the understanding of extreme polarization from social psychology emphasize the role of the group as a platform for an individual's transformation, where violent means to achieve shared group goals can become normalized[54,63]. In this research, the role of intragroup processes is often implied, but not often explicitly tested. That is, in most group-based models the constant activity occurring within groups, including micro-level processes such as spontaneous, random, and repeated intragroup interaction, has rarely been systematically studied. Thus, there is only limited research directly investigating the role of intragroup interaction on polarization.

Even in classic research focusing on intragroup processes, where polarization is captured by shifts in individual positions towards the extreme[29], interaction, mostly in the form of group discussion has been only studied using cross-sectional designs, as opposed to being observed or measured directly. The findings of these studies, including those on the effect of experimenter facilitated group discussion on extreme polarization[64,65], are important. However, these findings could be validated and extended by research that captures intragroup interaction as it occurs in natural contexts, that is, as a continuous and re-iterative process within a group using complex systems modeling approaches. Such an approach to the study of interaction would allow for key aspects such as the quantity and quality of intragroup interactions to be taken into consideration. Thus, by integrating group-based perspectives on interaction with the understanding of society as a complex system, we can, for example, obtain a clearer picture of the dynamics of human interaction within smaller networks, which in turn are nested within larger groups, or societies as a whole[65,66]. Taking the interactions between social actors within a broader network into consideration, we can model the drivers that qualify these interactions, that is, the rules of interaction, based on social psychological theory. By bringing together an understanding of society as a complex system and an understanding of identity and the behavioral dynamics of polarization from social psychology it is possible to gain a more nuanced understanding of the how societies gradually polarize, and ultimately cluster into cells having extreme beliefs.

## Social psychological processes driving polarization

Psychological groups, defined by their internal cohesion, in which members perceive themselves as pursuing promotively interdependent goals[67], are unified through a shared consensus on collective narratives about important aspects of social reality[19,26]. As such, the group represents a critical link between the individual and society, serving as both a shared cognitive representation of a collective entity and a collection of individuals who are united by a shared understanding of social reality. Self-categorization theory[62] sees polarization as increasing adherence to group norms, a process accentuated through interaction with outgroups. Within groups, collective norms are transformed through a process of consensual validation[68,69], but at the same time are also influenced by the broader intergroup context via interaction with relevant outgroups[70]. This means that stronger group identification, often in combination with other factors such as perceived outgroup threat, for example, tends to be a strong predictor of polarization[70,71], including in contexts of partisan polarization[72,73].

Classic research in social psychology reports polarization being heightened through internal group discussions that lead to more extreme positions[29]. More recently, this dynamic was studied in the context of extreme polarization, as for example in research on animal welfare activism where intentions to engage in radical action were strengthened by group discussion[64]. Well-researched drivers of collective action for social change, such as perceptions of group injustice and group threat, national identification, and relative deprivation, are also used to explain extreme polarization and radicalization[74], in addition to less studied factors such as group-based significance loss[75]. Group identity is vital in the study of these processes and is often conceptualized as a "need to belong". It is key to both the early and later phases of radicalization, providing the normative frame for sanctioning political violence[76]. In a similar vein, the quest for significance

model, for example, also emphasizes group-based aspects such as ideology and social processes within a dynamic radicalization process[77,78]. As previous research has shown, the role of the group is fundamental to our understanding of social dynamics. In the following section we extend this argument by examining the role of the group and its evolution as the society transforms to become more fragmented and the process of polarization unfolds.

## Polarization as the gradual fragmentation of society

We propose that polarization can be seen as a dynamic process underpinned by fragmentation at the level of the society, the group, and the individual, each having a distinct outcome. That is, dissent over an important societal issue triggers fragmentation, or division in society, which leads to the formation of opposing ideological camps. This is often evident when societies bi-polarize on an important issue. Fragmentation within the camps then leads to the formation of dissenting factions. When these factions become more extreme than the parent camp, they distance themselves (polarize) from both the initial opposing camp as well as the parent camp. Further fragmentation within already extreme factions may then lead to the formation of radical cells, which become increasingly ideologically distanced from the initial opposing camp, mainstream parent camp, and the parent faction. Thus, dissent triggers a transformation process of gradual fragmentation of the whole (a society or community) underpinned by increasing social identity differentiation and evolution of what "us versus them" represents.

We identify three distinct contexts where the process of fragmentation leads to different outcomes. First, at the level of a society (or unified community more broadly) fragmentation manifests as division into opposing camps. That is, ideological disagreement on societal issues (underpinned by contrastive narratives about social reality), leads to the division of society into ideologically opposed camps which in turn are consolidated by internal consensus[79], becoming polarized and segregated. Second, when there is dissent within camps, the process of fragmentation results in the formation of splinter factions that can further polarize and become more extreme than the parent camp. As these groups further polarize and segregate, including within online communities[72,73], they create an environment fostering intragroup conflict, potentially escalating polarization[29,74]. Finally, dissent within a faction, may lead to further fragmentation. Clustering within an already extreme subgroup can again lead to factionalization and identity fusion, where personal and group identities converge[80–83], and intense clustering (bonding) of extreme group members occurs. Thus, dissent in extreme factions, where views may be more extreme than the parent faction, can cause further extremization and even radicalization where members may justify the use of political violence to achieve group goals.

This process is illustrated in Fig. 1, which shows a simulated artificial society where a divisive issue was introduced. In this society, its members (agents) are modeled to hold a belief on an issue (the basis of opposed camp formation). This is shown as a gray scale with white and black representing extremely positive or negative views. Social connections are represented by links between agents. The figure shows that beginning from a disordered state, dissent and consensus through successive social interactions results in individuals re-evaluating their belief on an issue and forming and dissolving affiliations. This leads to the emergence of opposed camps, that divide into splinter factions and, eventually, extreme clusters.

The transformations in the simulated society can be seen as evolutionary stages when viewed sequentially. Structural inequalities and societal conflicts can pave the way for the formation of ideologically opposed camps, triggering a deductive (top-down) process that may lead to extreme polarization. However, this process is also inductive, as the transformation is ultimately driven by the members of the society as interacting agents. This transformation involves emergent division, intragroup fragmentation, and clustering into small extreme subgroups. Individuals, groups, and societal factors, as well as their interactions, determine the direction and speed of the transformation process, but individuals may move between different phases[84] (toward or away from extreme positions).

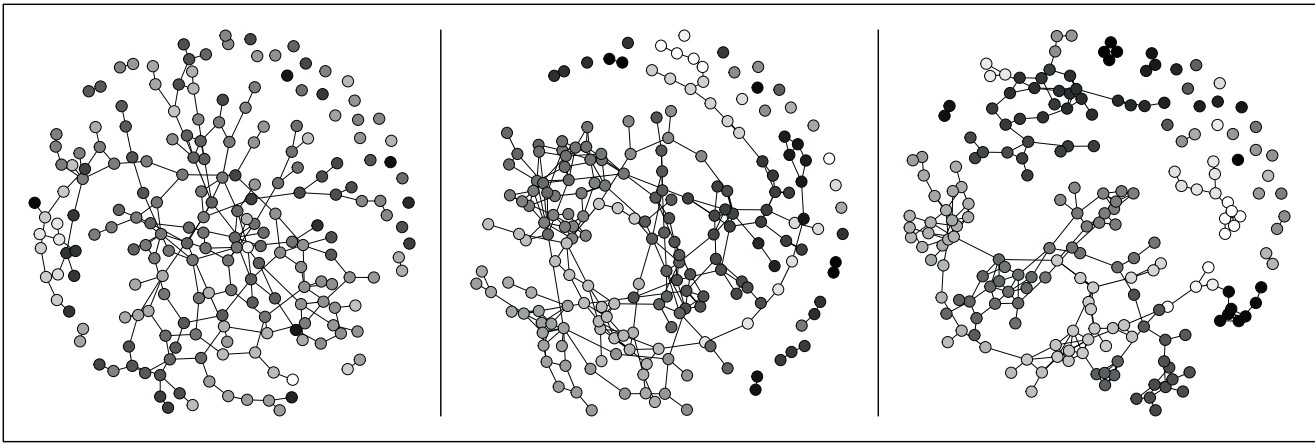

**Fig. 1 | Simulated evolution of an artificial society** The three panels illustrate the phases of polarization (from left to right): the formation of camps, factions, and radical cells.

Not entirely surprising, the triggers for this process are concomitant dissent and consensus. On the one hand, dissent in society about aspects of social reality creates a platform where conflicting narratives seeking to explain aspects of social reality can develop, leading to division into ideologically opposed camps (for example, Brexit supporters and opponents[85]). Dissent within camps may lead to the formation of splinter factions, with further dissent within these factions resulting in smaller clusters having extreme views, from which radical subgroups may form. On the other hand, consensus within camps, factions, and cells unifies them, ensuring their survival and functionality as psychological groups, thereby enabling collective processes that sustain the refinement and transformation of the group narrative.

As groups polarize and fragment, changes in narratives mirror the changes in the group members' shared beliefs, values, including moral convictions[86–88], and norms about group behavior[26]. That is, changes in the social identity content of the group[30,89]. To illustrate our argument, we consider how collective narratives may vary at different phases of transformation, using dissent about the value of cultural diversity in society as an example. For someone aligned with a center-right ideology, this may mean voting for an anti-immigration politician. Those identifying with a more extreme faction may support harsh immigration policies and enforcement methods (a more extreme narrative). At the furthest extreme, individuals may endorse or engage in violence against minority groups. This progression illustrates how increasingly radical narratives can lead to more extreme attitudes and behaviors.

The role of dissent in fracturing societies is clearly shown in the global warming debate, which has polarized the public into opposing ideological camps. Although, broadly speaking, there is scientific consensus on the anthropogenic causes of global warming, the public is divided into those self-identifying as climate change skeptics, in line with a narrative that contradicts most scientific evidence, and an ideological camp of people who support a narrative aligned to climate change science. Focusing on the supporters of mitigating climate change, there is fragmentation within emerging factions (some more extreme than others) which not only endorse pro-environmental behaviors and political action, but also engage in radical and sometimes violent action (labeled as "eco-terrorism" by some media, government organizations and the corporations targeted).

Our framework helps us understand (post-hoc) instances of extreme polarization and radicalization, including political violence. By examining how these processes developed in specific communities and socio-political conditions, we can design more effective preventive measures. Additionally, this framework can identify high-risk conditions for polarization and radicalization, enabling timely interventions before critical points are reached. While we do not maintain that dissent and polarization would always result in extreme outcomes, in divided societies, regardless of the end

result, the process of polarization is likely to develop asymmetrically, as even from the emergence of a divisive issue, the resulting camps would differ in their alignment to the dominant political power. This is important, as it will likely determine how active and engaged camp members are. For instance, the collective goals of an ideological camp aligned to the dominant political power would be in line with the status quo in a society (e.g., climate skeptics in a society governed by political conservatives supportive of the oil industry). For the opposed ideological group, going against the status quo (thus requiring more effort to produce social change) the processes of fragmentation and further clustering are likely to be more intense, resulting in more diverse social identity content. Without having power on their side, group members need to be more active and innovative in devising means to achieve group goals that challenge the status quo[90].

The concept of group identification is important here for several reasons. First, transformations in the collective narratives shared within a group are shaped by the unique context of intergroup and intragroup interactions in both online and offline social networks. However, these transformations are only possible if group members identify with a group and interact as group members. Secondly, in a similar process, the societal transformation of ideologically opposed camps into dissenting splinter factions, and ultimately extreme subgroups, is driven by intragroup and intergroup interactions between people identifying with camps, factions, and cells. Moreover, these social identities help create a context within which individual members of a society may interact as group members with agency, enabling them to be influential (and be influenced) in their capacity as either ingroup or outgroup members. Thus, mutual social influence and change can occur both within groups, driven by expected consensus with ingroup members in an intragroup context[57,58], and between groups, driven by expected dissent with outgroup members in an intergroup context. Table 1 shows the process of polarization underpinned by group fragmentation and changes to collective identity, as collective narratives are transformed within camps, splinter factions, and cells. The following section describes these transformations in more detail.

## Phases of societal fragmentation
### Ideologically opposed camp formation
Camps are groups based on shared narratives about social reality[64,91]. Conflict occurs due to alternative interpretations of important aspects of that reality[92,93]. These narratives are often aligned with opposing or mutually exclusive world views, moral values, and political orientation[94,95]. For example, conflicting narratives include those based on dissent about interpretations of history in intractable conflicts[96–98]. While conflicting narratives may divide societies into opposed camps, consensus within these camps, that is, self-identification with like-minded people[99], forms the basis of identity maintenance and development. This means that, within each

**Table 1 | Framework showing the process of polarization at varying levels of transformation (society, group, and individual) and in different contexts (camps, factions, and radical cells), resulting in different societal outcomes**

| | | Stage of Fragmentation | | | Key change mechanism |
|---|---|---|---|---|---|
| | | Camps | Factions | Cells | |
| Level of transformation | Society | Dissent within society leads to the formation of ideologically opposed camps | Dissent within camps leads to the formation of splinter factions | Dissent within factions leads to the formation of radical cells | Division and increasing differentiation |
| | Group | Camp narratives shape group norms and interaction within and between camps | More specific faction narratives shape group norms and interaction within faction and camp | Extreme cell narratives shape group norms and interaction within cell and between cell and faction | Gradual differentiation of social identity content within each group as contrast with outgroup increases |
| | Individual | Camp social identity content is internalized and refined | Faction social identity content is internalized and refined | Cell social identity content is internalized. Strong bonding and identity fusion may occur | Subjective experience of the individual in the group |
| Societal outcome | | Polarization between camps: DIVISION | Polarization within camps: FACTIONALIZATION | Polarization within factions: RADICALIZATION | |

opposing camp, the group narrative shapes the content of the collective identity with its associated values, beliefs, and norms.

From the perspective of group members, the strength of social identification reflects the importance of the group to the individual[100]. If we apply this argument to opposed camps, the strength of identification should be reflected in the degree of ideological certainty about the group narrative. That is, certainty about the group-defining narrative representing the truth from the perspective of the group. Of course, other sources of social identity strength are available to group members, including leadership, power struggles, and social cleavages. For members of ideologically opposed camps, however, the interactions between ingroup and outgroup members, by which group-defining narratives are validated or invalidated via discussions, are likely to affect the group members' levels of certainty. This creates the need for group members exposed to the outgroup's position to seek further validation from the ingroup. Conversely, it is likely that members of camps would seek to avoid exposure to narratives and arguments contradicting their own. For example, both committed liberals and conservatives prefer to stay in their own ideological bubble rather than be confronted with conflicting narratives on contentious social issues[101]. It follows that intragroup interactions are likely to become preferential as members of ideologically opposed camps would actively seek validation and re-endorsement of the ingroup narrative (confirmation bias) and avoid challenges from outgroups[4]. Thus, while group members would be attracted by interactions with ingroup members, they would be ideologically "repulsed" when interacting with outgroup members. This pattern of preferential interaction can lead to ideological segregation through exclusive communication with others who have the same point of view, as in "(…) liberals watching and reading only liberals; moderates, moderates; conservatives, conservatives; Neo-Nazi, Neo-Nazi"[102] (p.9), which can drive the further transformation of society, through further polarization between ideologically opposed camps. Within a camp where consensus is expected[58,60], interactions between group members validate and reinforce the camp identity, so that a process of mutual influence towards the crystallization of identity may occur[103].

This process of validation is reflected in increased certainty about the group's collective narrative[104], and therefore, results in strengthening identification with the respective camp. However, the alternative pathway of intergroup interaction is equally important. That is, when interactions between the ingroup and outgroup members occur, and disagreement is expected, exposure to the outgroup's alternative collective narrative can reinforce conviction in the group narrative with the associated values and norms that differentiate the ingroup from the outgroup, thereby increasing polarization[105]. Invalidation in this case could still lead to the same outcome of increased certainty in the ingroup position[106,107]. Thus, certainty about the group narrative can be strengthened and polarization increased via a dual path, both when the narrative is reinforced via intragroup interaction, and when the narrative is contrasted with that of the outgroup[108]. Thus, while people might prefer to interact online with people and media content reinforcing their view, social media platforms provide many opportunities to exposure to different and opposing views, much more so than would be possible in the real world[109]. However, intergroup interaction in these situations (exposure to outgroup's position) can also increase polarization, as exposure to opposing views can lead to an increase one's own belief conviction[110] rather than encouraging reflection and change.

### Splinter faction formation

Polarization between camps and the resulting ideological segregation in camps provide opportunities for intensive intragroup interactions and deliberation within camps, as shown in early experiments on group polarization[111]. Within a group that has many opportunities to deliberate, but limited exposure to outside information, the group position will likely become more extreme[60,62], and thus further distanced from the opposing camp. However, even in highly homogenous groups, intragroup dissent, for example in the form of disagreement with the norms and actions taken by the group, may develop. Splinter factions can form within opposed camps when group members disagree on fundamental issues of group identity and

the disagreement cannot be resolved within the group. In this situation, the shared reality of group members is compromised. Thus, splinter factions form via a process of schism, or intragroup fragmentation, as members distance themselves from the mainstream camp[112,113]. When this happens, the dissenting faction may develop different interpretations of the mainstream group narrative, and this changed social identity content is reflected in the adoption of new norms based on updated self-definitions of the identity of the group.

Those driving intragroup fragmentation are likely to be group members who strongly identify with the parent camp. As the normative conflict model[114,115] suggests, it is those members who identify more strongly with the group who are most likely to challenge group norms when they believe that current norms no longer serve the achievement of group goals. Individuals who are highly invested in the group are more likely to evaluate the norms (and what they mean for the group) and deviate from the group prototype, if this deviation is seen as being in the best interest for the group[114,116]. Once splinter factions within a camp are formed, the nature of the ingroup is reconfigured. Thus, from the perspective of faction members, their ingroup now consists of other members of the faction, while members of the parent camp become the outgroup, alongside members of the opposed camp. Thus, while the ingroup becomes smaller and more homogenous (more ideologically specified and narrow), the outgroup composed of both members of the opposing and parent camps, becomes more diverse, encompassing both different degrees of opposition and opposition to different aspects of the group's identity.

Whereas societal polarization into camps arises as an outcome of dissent between opposing ideological groups, polarization within a camp is characterized by a more extreme separation between ingroup and outgroups. Splinter factions are likely to consist of "high identifiers", members who are highly invested in the group, and high in certainty about the group's position. Validation from the ingroup and invalidation from the outgroup intragroup would most likely increase certainty about group position, which would, in turn, increase ideological segregation and polarization. We refer to the intense clustering of like-minded group members within factions as a heightened form of ideological segregation occurring between the splinter faction and parent camp.

### Radical cell formation

We argue that the formation of radical subgroups, or cells, that is, small groups composed of extreme and psychologically bonded members, arise as the result of intense clustering within a faction. Extreme clusters can form in physical spaces such as neighborhoods, but they often occupy online spaces that go beyond geopolitical boundaries. Either offline or online, they are highly cohesive psychological groups that are high in entitativity[117], so they can readily become platforms for identity fusion, a process whereby personal and group identity become one[80,81]. Because group membership is central to a person's self-definition, the boundaries between personal and collective self might disappear to be replaced by a "visceral sense of connection with the group"[79,82]. Understanding how these cells form is most important to political psychologists and policy makers since it is from these groups that political violence usually occurs.

The formation of these extreme clusters is similar to the intragroup fragmentation leading to factions. The process leading to the formation of radical subgroups is again characterized by intense ideological clustering, whereby the group intensifies its position through ingroup consensus and outgroup dissent[118]. Viewed as processes within a complex system, consensus underpins the formation and preservation of clusters, whereas dissent is the basis of fragmentation and alienation from other groups and mainstream society[119]. When splinter factions move away from the parent camp, they also become more polarized in their identity content, so that that their values and beliefs further diverge from those of the parent camp. Extreme clusters formed within these factions can develop identity content that is even more extreme than that of the splintering faction. This content is reflected in new (extremist) group norms that, for example, may sanction the use of political violence[120]. The growing isolation from outgroups, now consisting of not only the opposed camp, but also the parent camp and splinter factions, can make the group's narrative more powerful because group members find themselves without any other source of support or alternative group norms to adhere to[121]. Thus, identification with an extreme group should predict support for violent political action through the process of identity fusion.

A key contributing factor to the extremization of members in these radical clusters is minimal interaction outside their ingroup. Under these conditions, isolation from more moderate factions and the parent camp can lead to powerful group cohesion. For example, isolation in an underground cell will magnify the intensity of violence and justify the escalation of violent tactics. An illustration of this process is the powerful cohesion that develops in small combat groups when soldiers are separated from everyone except their comrades. Under these conditions, isolation and interdependence leads to extreme group cohesion. A bond described as being "closer than brothers"[121]. In intractable conflicts, intense interaction in online ideologically homogeneous groups was linked to the normalizing of dehumanization and hatred of the outgroup[28]. Alienation from an increasingly larger and heterogenous outgroup, and intense bonding in conjunction with identity fusion[81,82] can create the conditions under which extreme subgroups commit acts of political violence.

### Implications of the proposed framework

This review has presented an integrative framework for the study of polarization based on the view of society as a complex system made up of individual agents driven by evolving beliefs, values, and norms within increasingly refined and narrow social identities. The purpose of this framework is to illuminate the unique role dissent plays in the process of polarization at different levels of society – a process that can culminate in extreme polarization and radicalization. The framework can be applied to contexts where extreme polarization has already occurred, is emerging, or where there is a high risk of it happening in the future.

Within this framework, polarization is seen as a dynamic process driven by social interaction and manifested as the transformation of society from an heterogenous state (in which everyone can interact with everyone else) to a highly ordered state that can culminate in a clustering of extreme beliefs, once a divisive issue is introduced. Within this framework, polarization is a process that can be observed at the level of a society, but reflects the transformation of the group and the individual within the group. That is, this transformation takes the form of an internalized social identity change that culminates in the increased susceptibility of conforming to extreme group norms, along a path to radicalization.

Our framework also clarifies the roles of dissent and consensus in the dynamics of polarization. Dissent is inherent wherever human interaction is present, as any aspect of social reality entails a degree of ambiguity or space for alternative interpretations, leading to ideological diversity. As people interact, communicate, and deliberate with either ingroup or outgroup members, nuanced views of the collective self in the group emerge, leading to a further differentiation of narratives, providing a clear basis for expressions of dissent. Thus, while consensus represents an ordering force that brings together those holding similar views, dissent is the force that fragments and pulls people apart. These two forces work concurrently, pulling groups and group members in opposing directions, and underpinning group dynamics resulting from factors such as influence, repulsion, and attraction, among others. Our framework highlights the importance of the interplay between dissent and consensus as the basis of both diverging narratives at the intergroup level, and diverging elements of a same, shared intragroup narrative.

Computational models are being increasingly used to study radicalization driven by opinion dynamics and to identify the mechanisms responsible for initial fragmentation, segregation, and evolution to extreme opinions[53,54]. While some of these models effectively combine classic social identity and social influence theories with agent-based modeling[24,25], many fail to account for the critical impact of group dynamics and social identity on agent behaviors, as evident in recent contributions to the psychology of collective action and ideological polarization[63]. A contribution of our

current work is that it brings together social psychological analyses of group-based processes and computational model approaches showing polarization to be the emergent outcome of bonding based on belief and identity, driven by the actions of independent agents in a society.

We summarize our theoretical contribution as follows. First, our review articulates an integrative treatment of opinion, as a position towards a collective narrative, and identity, as group identification and self-definition as a supporter of that specific collective narrative. Thus, we extend the argument that the group represents a platform for extreme polarization, by specifying the type of groups that are most likely to polarize. Second, our framework extends current accounts of the process of polarization by including a pathway to radicalization that reflects the connection between radical subgroups, broader social movements and conflicting collective narratives circulating in societies. Dominant approaches to understanding political violence directly link behaviors to worldviews based on contrastive narratives. For example, in leaderless jihad, political violence by Western Muslims is often seen as resulting from views that divide the world into those waging war on Islam (proponents of the war on terrorism) and those defending Islam[122]. We advance this argument by proposing that support for contrastive narratives about social reality represents the beginning of a gradual process of polarization that can ultimately lead to radicalization.

By expanding on ideas of social identification and social identity content, our understanding of extreme polarization incorporates clearly specified processes and paths resulting from the interactions of group members (as members of ideologically opposed camps, splinter factions, and extreme clusters) to macro-level transformations at the level of society. This theoretical integration also helps bridge the disconnect between research on extreme subgroups and research on the individual, group, and mass polarization processes, providing psychological explanations of societal dynamics derived from complex systems science. When support for conflicting narratives in society constitutes the basis of collective self-definition and group formation, consensus and dissent become key forces shaping societal transformations.

As dynamic complex systems, societies are continually transforming, moving between disorder, in the form of dissent and fragmentation, and order, in the form of consensus and unification[118]. If extreme subgroups are understood as highly ordered systems (their members highly similar in belief conviction and group identification), their formation can be represented as a macro-level outcome of a state transition, typical of other complex systems in nature. In these systems, the elements of a disordered system evolve towards an ordered state, so that heterogenous and chaotic elements become more fixed, homogenous, and clustered. In this framework, the concepts of order (consensus and homogeneity) and disorder (dissent and heterogeneity) are applied to beliefs about society, which in turn can provide a basis for collective identity formation[92]. The dynamics that drive such societal outcomes form a continuous feedback loop. In other words, the actions of individual agents as group members change the structure of society, which in turn affects the future actions of agents. From our perspective, agents must be modeled as complex entities that can influence and be influenced by others, in similar ways to that proposed by dynamic social impact theory[50,51] but their actions are driven by distinctive social identities. It is through changes to individuals resulting from this mutual influence that profound social change can be achieved.

A practical implication of this work is that understanding polarization as a dynamic process driven by human interaction gives support to the use of social interventions to slow, interrupt, or remediate extreme polarization. For example, the use of social influence to attract people toward a more central view on issues[83], moderating communications within and between groups or introducing superordinate goals relevant to both sides of a conflicting issue are potential candidates for preventive programs or depolarizing interventions. These ideas could be tested via agent-based simulations and studies of communities, online or in the real world.

By modeling the transformation process we propose using real data, it may be possible to identify communities that might be at high risk of extreme polarization or about to reach a tipping point in their evolution (for example, from moderate camp to extreme faction). Thus, on one level, this framework can be further developed to identify the conditions for political extremism to develop, on another level, it can be applied to assess the effectiveness of current strategies to address extremism and political violence, or design new ones. For example, attempts by authorities to dismantle extreme clusters have been shown to be ineffective as new extreme clusters emerge from the remaining splinter group to fill the vacuum. This is because the structural conditions leading to polarization remain unchanged. By contrast, our approach suggests that by pairing the removal of extreme clusters from communities with interventions seeking to address the structural conditions in those communities as well as dysfunctional socio-psychological processes (such as limited social interaction), the re-emergence of extreme clusters may be prevented. Alternatively, prevention programs can target pre- or early polarization in vulnerable communities by supporting the formation of inter-community linkages with a focus on shared goals that are meaningful to group members in opposed camps.

## Conclusion

To date, there have been limited attempts to integrate the study of the individual and the society in understanding polarization and radicalization. Here we propose an integrative framework based on accounts of polarization as both reflected in the fragmentation of society and underpinned by the transformation of the individual and the group (as increased susceptibility of conforming to increasingly extreme group norms). Within this framework, the gradual evolution of polarization is seen as the outcome of a dynamic process resulting from dissent and the interplay of processes at the individual, group, and societal levels. Individual actions are driven by the social psychological processes of group identification and social identity change, whereas social interaction creates specific contexts with opportunities for the refinement and transformation of social identity content. Intergroup and intragroup interactions bridge these transformations in individuals, groups, and society, creating an effective framework to understand structural change in society through the actions of individual social agents.

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

## Author contributions

A-.M.B.: Conceptualization, Writing—Original draft preparation, Reviewing and Editing. J.M.B.: Conceptualization, Writing—Reviewing and Editing, Visualization. M.V., A.B., M.C.: Writing—Reviewing and Editing.

## Competing interests

The authors declare no competing interests.
