## [Peer Review File · Communications Psychology]

18th Aug 23

Dear Dr Bliuc,

Thank you for your patience during the peer-review process. Your manuscript titled "From Ideologically Opposed Camps to Extreme Clusters —A Narrative Review of Polarisation in Divided Societies" has now been seen by 2 reviewers, and I include their comments at the end of this message.

The reviewers are in principle enthusiastic about your work. However, they also mention a number of concerns. We are very interested in the possibility of publishing your manuscript in Communications Psychology, but would like to consider your response to these concerns in the form of a revised manuscript before we make a decision on publication.

In detail, we ask you to address Reviewer #1's concerns calling for a stronger conceptual grounding, clear definitions, and explanations for real-world phenomena as well as transparency about where the framework does or does not likely yield reliable predictions. Moreover, Reviewer #2 offered constructive comments on how the work may better incorporate existing literature and more faithfully present existing work, which we likewise ask you to incorporate.

In sum, we invite you to revise your manuscript taking into account all reviewer and editor comments.

EDITORIAL POLICIES AND FORMATTING

You will find a complete list of formatting requirements following this link:

<https://www.nature.com/documents/commsj-style-formatting-checklist-review-perspective.pdf>

Please use the checklist to prepare your manuscript for resubmission.

* **TRANSPARENT PEER REVIEW:** Communications Psychology uses a transparent peer review system. This means that we publish the editorial decision letters including Reviewers' comments to the authors and the author rebuttal letters online as a supplementary peer review file. We publish these records for all accepted manuscripts. However, on author request, confidential information

and data can be removed from the published reviewer reports and rebuttal letters prior to publication. If your manuscript has been previously reviewed at another journal, those Reviewers' comments would not form part of the published peer review file.

If you have any questions about any of our policies or formatting, please don't hesitate to contact me.

Please use the following link to submit your revised manuscript and a point-by-point response to the referees' comments (which should be in a separate document to any cover letter):

[link redacted]

We hope to receive your revised paper within 12 weeks; please let us know if you aren't able to submit it within this time so that we can discuss how best to proceed. If we don't hear from you, and the revision process takes significantly longer, we may close your file.

We understand that due to the current global situation, the time required for revision may be longer than usual. We would appreciate it if you could keep us informed about an estimated timescale for resubmission, to facilitate our planning. Of course, if you are unable to estimate, we are happy to accommodate necessary extensions nevertheless.

Please do not hesitate to contact me if you have any questions or would like to discuss these revisions further. We look forward to seeing the revised manuscript and thank you for the opportunity to review your work.

Best regards,

Marike

Marike Schiffer, PhD

Chief Editor

Communications Psychology

REVIEWERS' EXPERTISE:

Both referees work at the intersection of political science, psychology, and computational social science.

REVIEWERS' COMMENTS:

Reviewer #1 (Remarks to the Author):

I enjoyed reading this timely manuscript presenting a framework integrating social psychology and computational social science understandings of polarization. The manuscript is well written and concise, which I appreciate, and is attempting to integrate streams of research which are indeed in need of theoretical integration. When I read pieces like this, I do so with two criteria in mind. The first is whether the framework presented is internally coherent. Does it present a consistent model of causal processes, are its constituent theoretical elements clearly defined, etcetera. The second is usefulness, does the framework provide scholars researching the topic an avenue for generating novel hypothesis, does it elucidate previously unexamined phenomena, etcetera. For both criteria, I find the manuscript, as written, insufficient. Below I provide comments and suggestions in the hopes that they are helpful to the authors as they refine this work.

Allow me to describe, in my own terms, the central argument of the manuscript. Polarization (issue-position extremity) can be understood as a process by which groups disagree over symbolic issues (“dissent” over “collective narratives”) which results in recursive identity-group fracturing, at every level of which the groups become more extreme in their issue positions, identification, and support for violence.

My critique of this framework is that (1) it is descriptive, not explanatory, and (2) its elements are poorly defined. The idea of “dissent” as a central mechanism is an acute example of these two criticisms. Dissent is not defined in the manuscript. What I believe it means is just disagreement, but it’s used in different contexts and “Stages” of the model to mean different things. In Stage 1, it represents the type of disagreement that’s analogous to disagreement between Democrats and

Republican, i.e. core differences in values. But then in Stages 2 and 3 (which to me are indistinguishable) “dissent” means intra-group conflict, analogous to (for the sake of example) disagreement between social conservatives and economics conservatives (two factions in the Republican Party) in Stage 2, and analogous to (to extend this example) far-right militias and social conservatives (the former of which is a more extreme faction within the latter). The relationship, both in terms of substantive (dis)agreement and identity, between all these groups is not the same. The “dissent” between far-right militias and the social conservatives is not the same as the “dissent” between the median Democrat and median Republican, but in this model they’re treated as equivalent. This is also true for how the manuscript treats identity.

That equivalency would be an acceptable abstraction if the model provided a framework for understanding and predicting when a group does or does not experience internal dissent and fracturing, or for examining relative levels of dissent and fracturing, but it does not. In Stage 1, echo chambers lead to identity cohesion, but then in Stages 2 and 3 this somehow flips and echo chambers lead to identity fracturing. The preface to Stage 2 merely states that “intragroup dissent may develop.” As a researcher of polarization, isn’t *that* the phenomenon we want to study, *when* dissent arises versus when it does not? The model, as presented, offers no explanation or when, where, or why dissent arises, it’s just a given. This is what I mean when I say the model is descriptive, rather than explanatory. The model needs to articulate a clearer definition of dissent, and incorporate elements that explain when and why dissent arises and when consensus arises.

This leads to another critique I have of the model: to what real-world examples of conflict does this model apply? Taken at face values, this model would suggest all conflict inevitably leads to maximal balkanization (for want of a better term), yet obviously not all intergroup and intragroup conflict leads to violent extremist splinter groups arising. The model is presented as a general model of polarization, yet what it characterizes does not seem representative of most intergroup conflict in modern democracies. This manuscript needs to more narrowly locate the specific real-world outcomes this framework is attempting to explain.

The model also does not sufficiently incorporate past literature. On pg. 6 it states, “That is, in most group-based models the constant activity occurring within groups (micro-level processes such as spontaneous, random, and repeated intra-group interaction) has rarely been systematically studied, with limited research directly investigating the role of intragroup interaction on polarization.” Yet the very term “group polarization” was coined to describe how intragroup processes lead to more extreme group attitudes (Moscovici & Zavalloni, 1969, JPSP). There are decades worth of studies precisely identifying the intragroup processes that lead to issue position extremity. What the manuscript describes as “Stage 1” resembles the classic group polarization theoretical account, but as I mentioned above the manuscript then completely inverts this model for Stages 2 and 3 where all the processes that lead to cohesion in Stage 1 produce fracturing. The manuscript needs to better incorporate this past work on group polarization, and clearly articulate what shifts between Stages 1 and 2-3 to explain the loss of group cohesion.

I'll end on a personal note. I identify as a researcher who does (some) computational social science (CSS). I think these methods have the potential to be revolutionary to social science research, and I believe the authors are right to perceive the theoretical integration of CSS and social psychology as a highly fruitful and generative pursuit. But I see CSS as a method for describing complex systems, I do not see CSS (at least as currently instantiated) as a discipline with specific theories of human behavior (unlike social psychology). When I read the introduction to this paper, I was surprised that they were treated as equivalent in this regard. I know I'm speculating here, but I suspect this equivalence is one reason why the manuscript ends up being more descriptive than mechanistically explanatory. CSS does not bring with it explanatory theories, which makes integrating it with social psychology in a manner providing greater explanatory value above what social psychology already does difficult.

I hope the authors find my comments helpful as they continue to refine this manuscript.

Signed Review: Jeff Lees

Reviewer #2 (Remarks to the Author):

This paper carries out a review of research across social psychology and computational social science, seeking to develop a comprehensive conceptual model of polarization which integrate both individuals, groups and society.

While I find myself disagreeing with some of the points that the authors make, I do find the paper interesting and I believe it warrants publications. The theoretical framework provided is a useful contribution, and other scholars may engage with it and test it empirically. If it is wrong, it is likely to be usefully wrong - which is perhaps the best we can hope for in science.

In the following review, I will suggest some points that I hope will further strengthen the paper, and in particular by suggesting some relevant additional literature with which they may want to engage. (However, the authors do not need to feel obligated to engage with every reference provided, if they do not believe that doing so will improve the paper.)

First, my main issue with the paper is that it makes some strong assumptions, and states as certain issues that are very much debated, or even highly controversial. I think that this is mostly an issue of the language used in referencing previous work. For instance, the paper starts with proclaiming the existence of a causal link between social media and polarization - which is quite a claim to make. It then proceeds to making assumptions about the supposed role of echo chambers in this process, which is even more controversial in the face of the current literature. The paper furthermore makes various assumptions about the nature of polarization, and so on. In my view, it is generally fine to make strong assumptions and to take minority views on issues, but it must be done carefully and making sure to not present as well-established views that are hotly debated. I would urge the author(s) to carefully go through their manuscript and nuance their statements, making explicit their assumptions, and more accurately and carefully represent the state of the literature. In many cases, this might consist of just adding a well-chosen "some scholars have argued", or stating "we will assume that".

One of the examples here that most bothered me was that the paper leans fairly heavily on the notion of echo chambers, and the classic Sunsteinian "homogeneity breeds extremism" assumption. From where I sit, this is now a minority view within the literature, as filter bubbles and echo chambers seem less prevalent than initially believed, and the causal link has been questioned by empirical work. I would like the authors to engage with this debate more explicitly. See e.g.:

Bruns, A. (2019). Are filter bubbles real?. John Wiley & Sons.

Bail, C. (2022). Breaking the social media prism: How to make our platforms less polarizing. Princeton University Press.

Second, a key issue with the paper is that it focuses on political polarization without engaging sufficiently with the literature in political science, the primary locus of debates around polarization and its drivers. Specifically, I believe this paper could be strengthened significantly by explicitly engaging with the ideas in affective polarization literature, and to better discuss the nature and foundation of polarization. This literature very explicitly builds on social psychology, and particularly social identity theory, and thus overlaps substantially with the aims of the paper at hand. The literature also has a significant focus on the relationship between identities and opinions.

The paper currently makes various more or less explicit assumptions about the relationship between opinions and identities (e.g., opinions -> identities). I would suggest the author(s) to engage with the affective polarization literature, and see how and whether they would like to incorporate it, or perhaps nuance their claims and assumptions.

The author(s) may find that the paper could become significantly stronger if it was theoretically situated within e.g., the affective polarization literature, allowing its contribution to be narrowed and sharpened.

Here are some examples of recommended reading:

Iyengar, S., Lelkes, Y., Levendusky, M., Malhotra, N., & Westwood, S. J. (2019). The origins and consequences of affective polarization in the United States. *Annual review of political science*, 22, 129-146.

Mason, L. (2018). *Uncivil agreement: How politics became our identity*. University of Chicago Press.

Achen, C., & Bartels, L. (2017). *Democracy for realists: Why elections do not produce responsive government*. Princeton University Press.

Robison, J., & Moskowitz, R. L. (2019). The group basis of partisan affective polarization. *The Journal of Politics*, 81(3), 1075-1079.

Webster, S. W., & Abramowitz, A. I. (2017). The ideological foundations of affective polarization in the US electorate. *American Politics Research*, 45(4), 621-647.

Druckman, J. N., Klar, S., Krupnikov, Y., Levendusky, M., & Ryan, J. B. (2021). How affective polarization shapes Americans' political beliefs: A study of response to the COVID-19 pandemic. *Journal of Experimental Political Science*, 8(3), 223-234.

Third, the author(s) argue for the incorporation of group identities in agent-based models, and drawing on social psychology. This is a good point, but it has to some degree already been made, as there is an ABM literature in CSS focused on the role of identities in polarization, and the relationship between opinions and identities. This work argues explicitly for the need to draw on social psychology when modeling opinion dynamics, for instance in the work of Tornberg (whom you cite in other contexts):

Törnberg, P., Andersson, C., Lindgren, K., & Banisch, S. (2021). Modeling the emergence of affective polarization in the social media society. *PLoS One*, 16(10), e0258259.

Törnberg, P. (2022). How digital media drive affective polarization through partisan sorting. *Proceedings of the National Academy of Sciences*, 119(42), e2207159119.

Fourth, a more minor issue: Figure 1 is said to come from a simulation, but no details are given. I would suggest either including a full description of the simulation, or not including the figure.

Reviewers' comments and our responses

Reviewer's #1 comments

General feedback: "I enjoyed reading this timely manuscript presenting a framework integrating social psychology and computational social science understandings of polarization. The manuscript is well written and concise, which I appreciate, and is attempting to integrate streams of research which are indeed in need of theoretical integration. When I read pieces like this, I do so with two criteria in mind. The first is whether the framework presented is internally coherent. Does it present a consistent model of causal processes, are its constituent theoretical elements clearly defined, etcetera. The second is usefulness, does the framework provide scholars researching the topic an avenue for generating novel hypothesis, does it elucidate previously unexamined phenomena, etcetera. For both criteria, I find the manuscript, as written, insufficient. Below I provide comments and suggestions in the hopes that they are helpful to the authors as they refine this work. Allow me to describe, in my own terms, the central argument of the manuscript. Polarization (issue-position extremity) can be understood as a process by which groups disagree over symbolic issues ("dissent" over "collective narratives") which results in recursive identity-group fracturing, at every level of which the groups become more extreme in their issue positions, identification, and support for violence."

Our response: We thank the reviewer for the positive assessment of our work as well as the detailed and constructive feedback. We appreciate the systematic approach to the review, which we believe significantly helped us refine and further develop some key aspects of our argument. Some of the reviewer's comments suggest that several of our most important points were not very clearly made, so to remedy this we re-wrote key parts of the paper. Below, we explain in detail how we integrated all the points raised by the reviewer into the revised manuscript.

Comment 1: General critique of the proposed framework. "My critique of this framework is that (1) it is descriptive, not explanatory, and (2) its elements are poorly defined. The idea of "dissent" as a central mechanism is an acute example of these two criticisms. Dissent is not defined in the manuscript. What I believe it means is just disagreement, but it's used in different contexts and "Stages" of the model to mean different things. In Stage 1, it represents the type of disagreement that's analogous to disagreement between Democrats and Republican, i.e., core differences in values. But then in Stages 2 and 3 (which to me are indistinguishable) "dissent" means intra-group conflict, analogous to (for the sake of example) disagreement between social conservatives and economics conservatives (two factions in the Republican Party) in Stage 2, and analogous to (to extend this example) far-right militias and social conservatives (the former of which is a more extreme faction within the latter). The relationship, both in terms of substantive (dis)agreement and identity, between all these groups is not the same. The "dissent" between far-right militias and the social conservatives is not the same as the "dissent" between the median Democrat and median Republican, but in this model they're treated as equivalent. This is also true for how the manuscript treats identity."

Our response: The comment on dissent is very important for the whole argument, so we start here by clarifying how we see the role of dissent within the proposed framework. The reviewer is right in the observation that dissent (consistently represented as *disagreement*

between positions/viewpoints) would work differently in different contexts and its outcomes would differ according to these contexts. We now explain in the paper that: a) initially dissent within a society (or community) would lead to the formation of opposing camps (broad bi-polarisation), b) then, dissent within camps would lead to the formation of factions (further polarisation), and c) finally, dissent within already extreme factions would lead to the formation of radical cells (extreme polarisation).

We also explain that, according to the context of dissent, its outcome can be *intergroup conflict* (in a society divided into opposing camps), or *intragroup conflict* (in a camp that fragments into factions, or in a faction fragmenting into cells/smaller subgroups). We have revised all the references we made to dissent throughout the paper to ensure consistency.

In describing how we addressed the remaining concerns of this reviewer we structure our response into three sections:

1) The framework is descriptive rather than explanatory (please also see our response to the point about predictions based on the framework under point 2). This criticism indicates that in our initial manuscript, we focused more heavily on the descriptive aspects of our model, neglecting to sufficiently emphasise its explanatory and predictive aspects. In the revised manuscript we rectify this by: a) revising the definitions of the main constructs for clarity and consistency (see also point 2 below), b) clarifying and emphasising the points on the explanatory and predictive value of the model, and c) including a clear set of predictions derived from the model (see also point 2 below on the specific predictions that we can derive from the framework).

2) Some elements of the framework are not defined well: dissent and identity

We now explain in the manuscript that, while the basic meaning of dissent as *disagreement between positions/viewpoints* is the same in all the stages of the model, there is an evolution of what dissent entails in three stages of the model as identities and associated identity content change. Thus, as the reviewer also observes, at the start of the process, dissent represents difference in position, as reflected in cases where something significant happens (e.g., Russia's invasion of Ukraine) and the public opinion in a European country becomes divided over support for or opposition to the official country's response, for example). As people start identifying with the specific narratives associated to support or opposition to the response (broadly categorised as pro-Ukraine versus pro-Russia), dissent is reflected as not only (broad) difference in position but also as further differentiation in terms of various aspects of identity content. This process of (further) differentiation is now occurring with the group – so disagreement about group values, beliefs, norms can lead to differentiation within a group – in addition to the differentiation in relation to the opposing outgroup. We now clearly explain how we define the central concept of dissent on page 2:

At the core of our proposed framework is the dynamic interplay between individual and collective processes within a divided society when there is *dissent* over a contentious issue. Dissent is defined here as disagreement on a position on an issue which is sufficiently important to provide a basis for social identity formation [21, 22], as for example, in relation to climate change.

The same argument applies to the *concept of identity*. One of the central ideas to our propositions is that group identity changes as the process of polarisation progresses. How identity is defined does never change (i.e., the mechanisms underpinning social identity

formation and development), but the *content* of the identity changes – for example, the identity content of opposing camps is likely to be different in significant ways from the identity content of splinter factions and radicalised cells. A good illustration of this point is how a national identity can have different, even contrastive beliefs and norms associated with such an identity – e.g., in the case of American national identity, one version of the identity can have gun ownership at its core while an alternative version of the same identity can have other central values (such as gun control as a mean to protect American citizens).

3) Distinctions between Stage 2 and Stage 3

Reflecting on the feedback, we realised that the way in which we referred to distinct stages to describe the process of polarisation was not adequately capturing the idea of polarisation as complex and progressive transformation engaging multiple levels of analysis. As a result, we revised how we talk about the process of polarisation — as being underpinned by three mechanisms with distinctive outcomes (formation of opposing camps, splinter factions and radical cells), rather than by distinct stages. For example, on page 4 we now say:

Polarisation has been generally regarded as a continuous process that, once started, and left unchecked, progresses towards increased distancing between opposing sides and ultimately full segregation [37]. That is, the repeated application of the mechanisms of ideological and affective distancing, leading to fragmentation into groups having more extreme positions. Conceptually, this process continues until a limiting state of complete segregation between the polarised groups is reached. To understand how to slow down or stop this process, we need to approach polarisation as a complex and progressive process driven by many interconnected factors, with crucial mechanisms of transformation within both society and individuals. As such, we identify three key mechanisms underpinning polarisation, which entail distinct transformations and outcomes in the society or group concerned, each broadly representing a more fragmented version of the preceding state. These mechanisms are: (a) division into opposing ideological camps; (b) fragmentation within the camps into dissenting factions; and (c) extreme clustering within factions.

Comment 2: Clearer articulation of what dissent is and what its roles are in polarisation. “That equivalency would be an acceptable abstraction if the model provided a framework for understanding and predicting when a group does or does not experience internal dissent and fracturing, or for examining relative levels of dissent and fracturing, but it does not. In Stage 1, echo chambers lead to identity cohesion, but then in Stages 2 and 3 this somehow flips and echo chambers lead to identity fracturing. The preface to Stage 2 merely states that “intragroup dissent may develop.” As a researcher of polarization, isn’t *that* the phenomenon we want to study, *when* dissent arises versus when it does not? The model, as presented, offers no explanation or when, where, or why dissent arises, it’s just a given. This is what I mean when I say the model is descriptive, rather than explanatory. The model needs to articulate a clearer definition of dissent, and incorporate elements that explain when and why dissent arises and when consensus arises.”

Our response: We have now further developed our argument on how the proposed framework can be used for hypothesis testing. Our predictions are framed to address the role of dissent on the process of polarisation and to further address potential ways to counteract polarisation. As such, we now include a discussion of potential predictions/testable hypotheses derived from our framework in the Discussion section on page 23:

Our proposed framework can form the basis of testable hypotheses about the polarisation process, including ways to counteract polarisation or ameliorate its worst effects. For example, we would expect that dissent on important societal issues will likely divide societies or communities into opposing ideological groups (camps), and that contrasting group norms will lead to broad polarisation between the opposing camps. Similarly, one would expect that dissent within a camp will lead to the formation of splinter factions – again different in identity content from the parent camp. Depending on the new social identity content, and in particular, the norms of the faction (which could be more extreme but also more moderate than those of the parent camp), the behaviour of group members will be either more extreme or more moderate (i.e., when factions split around the mean belief of the parent camp, inevitably one will be more extreme and the other more moderate, in the latter case the faction members either re-aligning with the mainstream parent camp or becoming unattached neutral “free agents”). Finally, dissent within a faction which is already more extreme than the parent camp is likely to lead to the formation of radical cells, again different in social identity content from the parent faction. In line with the refined/new group norms and unique to these small sub-groups, the behaviour of members will be characterised by extreme bonding and identity fusion.

In addition, we have now included a more focused discussion on practical implications. We now say on page 24:

In terms of possible practical implications of this work, understanding polarisation as a dynamic process driven by human interaction introduces the possibility of social interventions to slow, interrupt or remediate polarisation. Using social influence to attract people toward a more central view on issues [90], moderating communications within and between groups or introducing superordinate goals relevant to both sides of a conflicting issue could all work as part of preventive programmes or depolarising interventions. These ideas can be tested via agent-based simulations and through studies of real online communities. The findings of these studies will be crucial for inform social interventions and policies that aim to reduce the polarising effect of social identities promoting division in societies.

To address the point on **explanations of when, where, or why dissent arises**, we have included more detail on foundations and mechanisms of dissent. In addition to the definition of dissent (see above), we now have a discussion on how dissent functions at intragroup and intergroup levels. On pages 20-21, we now say:

Our framework also clarifies the roles of dissent and consensus in the dynamics of polarisation. Dissent is inherent wherever human interaction is present. As any aspect of social reality entails a degree of ambiguity or space for alternative interpretations, leading to ideological diversity. As people interact, communicate, and deliberate with either ingroup or outgroup members, nuanced views of the collective self in the group emerge, leading to a further differentiation of narratives, providing a clear basis for expressions of dissent. Thus, while consensus represents an ordering force that brings together those holding similar views, dissent is the force that fragments and pulls people apart. These two forces work concurrently, pulling groups and group members in opposing directions, and underpinning group dynamics resulting from factors such as influence, repulsion, and attraction, among

others. Our framework highlights the importance of the interplay between dissent and consensus as the basis of both diverging narratives at the intergroup level, and diverging elements of a same, shared intragroup narrative.

Comment 3: This leads to another critique I have of the model: to what real-world examples of conflict does this model apply? Taken at face values, this model would suggest all conflict inevitably leads to maximal balkanization (for want of a better term), yet obviously not all intergroup and intragroup conflict leads to violent extremist splinter groups arising. The model is presented as a general model of polarization, yet what it characterizes does not seem representative of most intergroup conflict in modern democracies. This manuscript needs to more narrowly locate the specific real-world outcomes this framework is attempting to explain.

Our response: In the paper, we say that the processes we describe do not have to always result in extreme polarisation and radicalisation. In particular, fragmentation within camps can lead to more moderate factions to form which would either re-align with the parent group or further fragment in moderate subgroups or “unattached” group members (“free agents”). However, our framework is meant to be used when problematic outcomes such as extreme polarisation and radicalisation do occur. This point is illustrated on page 24 where we say:

For example, we would expect that dissent on important societal issues will likely divide societies or communities into opposing ideological groups (camps), and that contrasting group norms will lead to broad polarisation between the opposing camps. Similarly, one would expect that dissent within a camp will lead to the formation of splinter factions – again different in identity content from the parent camp. Depending on the new social identity content, and in particular, the norms of the faction (which could be more extreme but also more moderate than those of the parent camp), the behaviour of group members will be either more extreme or more moderate (i.e., when factions split around the mean belief of the parent camp, inevitably one will be more extreme and the other more moderate, in the latter case the faction members either re-aligning with the mainstream parent camp or becoming unattached neutral “free agents”).

Comment 4: The model also does not sufficiently incorporate past literature. On pg. 6 it states, “That is, in most group-based models the constant activity occurring within groups (micro-level processes such as spontaneous, random, and repeated intra-group interaction) has rarely been systematically studied, with limited research directly investigating the role of intragroup interaction on polarization.” Yet the very term “group polarization” was coined to describe how intragroup processes lead to more extreme group attitudes (Moscovici & Zavalloni, 1969, JPSP). There are decades worth of studies precisely identifying the intragroup processes that lead to issue position extremity. What the manuscript describes as “Stage 1” resembles the classic group polarization theoretical account, but as I mentioned above the manuscript then completely inverts this model for Stages 2 and 3 where all the processes that lead to cohesion in Stage 1 produce fracturing. The manuscript needs to better incorporate this past work on group polarization, and clearly articulate what shifts between Stages 1 and 2-3 to explain the loss of group cohesion.

Our response: We now more clearly explain that our argument integrates the idea of group polarisation as an intragroup process in line with Moscovici & Zavalloni (1969). For example, we say early in the Introduction on page 3 (citing Moscovici & Zavalloni (1969)/33):

An important point here is that we see polarisation as having a dual pathway, being a process driven by both intergroup and intragroup interaction [33]. Fundamental to understanding this process is the recognition that polarisation is an ongoing and transformative process in which collective (as well as individual) identities are in continuous evolution over time [41].

Further on page 10 we say:

As these groups further polarise and segregate, including within online communities [79-79], they create an environment fostering intragroup conflict, potentially escalating polarisation [33, 80].

As also mentioned under our response to Comment 1 point 3, to better convey the complex and dynamic nature of polarisation, we have revised our framing of the argument so rather than referring to stages of polarisation (implying a clear temporal start and end and linearity of the process), we refer to mechanisms and outcomes of polarisation. We believe that this change also helps to clarify the distinctions between those. We have also changed Figure 2 in the paper to clarify these distinctions on page 14:

		Context of Fragmentation			Social Process
		Camps	Factions	Cells	
Level of Transformation	Society	Dissent within society leads to the formation of ideologically opposed camps	Dissent within camps leads to the formation of splinter factions	Dissent within factions leads to the formation of radical cells	Division and increasing differentiation
	Group	Camp narratives shape group norms and interaction within and between camps	More specific faction narratives shape group norms and interaction within faction and between faction and camp	Extreme cell narratives shape group norms and interaction within cell and between cell and faction	Increasing differentiation of social identity content within each group as contrast with outgroup increases
	Individual	Camp social identity content is internalised and refined	Faction social identity content is internalised and refined	Cell social identity content is internalised. Strong bonding and identity fusion may occur	Subjective experience of the individual in the group
Societal Outcome		Polarisation between camps: DIVISION	Polarisation within camps: FACTIONALISATION	Polarisation within factions: RADICALISATION	

Figure 2. Framework showing how polarisation occurs at varying levels of transformation (society, group, and individual) and in different contexts (formation of camps, factions, and extreme cells), resulting in different outcomes.

Comment 5: Relationship between social psychology and computational social science. “I’ll end on a personal note. I identify as a researcher who does (some) computational social science (CSS). I think these methods have the potential to be revolutionary to social science research, and I believe the authors are right to perceive the

theoretical integration of CSS and social psychology as a highly fruitful and generative pursuit. But I see CSS as a method for describing complex systems, I do not see CSS (at least as currently instantiated) as a discipline with specific theories of human behavior (unlike social psychology). When I read the introduction to this paper, I was surprised that they were treated as equivalent in this regard. I know I'm speculating here, but I suspect this equivalence is one reason why the manuscript ends up being more descriptive than mechanistically explanatory. CSS does not bring with it explanatory theories, which makes integrating it with social psychology in a manner providing greater explanatory value above what social psychology already does difficult."

Our response: We fully agree with the view of the reviewer on the relationship between social and political psychology and CSS. We have revised the whole manuscript to ensure that how we discuss the interplay between social/political psychology and CSS is representative of this view.

Reviewer's #2 comments

General feedback: "This paper carries out a review of research across social psychology and computational social science, seeking to develop a comprehensive conceptual model of polarization which integrate both individuals, groups and society. While I find myself disagreeing with some of the points that the authors make, I do find the paper interesting and I believe it warrants publications. The theoretical framework provided is a useful contribution, and other scholars may engage with it and test it empirically. If it is wrong, it is likely to be usefully wrong - which is perhaps the best we can hope for in science. In the following review, I will suggest some points that I hope will further strengthen the paper, and in particular by suggesting some relevant additional literature with which they may want to engage. (However, the authors do not need to feel obligated to engage with every reference provided, if they do not believe that doing so will improve the paper.)"

Our response: We thank the reviewer for the very constructive feedback, and we describe below how we addressed all the points raised in the revised manuscript.

Comment 1: Language use suggests too strong assumptions being made. "First, my main issue with the paper is that it makes some strong assumptions, and states as certain issues that are very much debated, or even highly controversial. I think that this is mostly an issue of the language used in referencing previous work. For instance, the paper starts with proclaiming the existence of a causal link between social media and polarization - which is quite a claim to make. It then proceeds to making assumptions about the supposed role of echo chambers in this process, which is even more controversial in the face of the current literature."

Our response: We have now revised the whole manuscript targeting the parts where the language is unnecessary strong and more nuanced needs to be introduced. We have also revised references to echo-chambers to reflect a more nuanced argument in line with some of the suggested work by the reviewer. For example, on page 1 we now say:

Problematic online encounters between both like-minded people and people with opposing views were shown to result in the reinforcement of extreme views and intensification of belief polarisation, reducing the willingness of groups to engage in constructive dialogue [3-7]. In extreme cases, this can lead to radicalisation, where individuals or groups adopt extreme views that sanction political violence [8-10].

Comment 2: Adding more nuance to the language used. “The paper furthermore makes various assumptions about the nature of polarization, and so on. In my view, it is generally fine to make strong assumptions and to take minority views on issues, but it must be done carefully and making sure to not present as well-established views that are hotly debated. I would urge the author(s) to carefully go through their manuscript and nuance their statements, making explicit their assumptions, and more accurately and carefully represent the state of the literature. In many cases, this might consist of just adding a well-chosen "some scholars have argued", or stating "we will assume that".

Our response: This is a very helpful comment, we have now revised the whole manuscript and changed our language in line with this recommendation. Some examples of how we implemented these changes include:

Consistent with this perspective, **we propose** that polarisation can be understood as the result of an iterative process of social interaction resulting in beliefs converging and diverging, and the making and breaking of social connections, that culminates in the clustering of extreme beliefs within a society. (p.5)

In modelling polarisation, including clustering and extremization, **we argue** that opinion is often treated as an entity that is disconnected from the complex social and personal identity of the agent. (p.7)

Comment 3: Echo-chambers argument. “One of the examples here that most bothered me was that the paper leans fairly heavily on the notion of echo chambers, and the classic Sunsteinian "homogeneity breeds extremism" assumption. From where I sit, this is now a minority view within the literature, as filter bubbles and echo chambers seem less prevalent than initially believed, and the causal link has been questioned by empirical work. I would like the authors to engage with this debate more explicitly. See e.g.:

Bruns, A. (2019). Are filter bubbles real?. John Wiley & Sons.

Bail, C. (2022). Breaking the social media prism: How to make our platforms less polarizing. Princeton University Press.

Our response: We agree with the reviewer’s more nuanced view on echo-chambers’ role in polarisation, in line with Bruns (2019) and Bail (2022), but clearly, the way in which we initially discussed this in the paper did not accurately capture this view. To remedy this, we now made several changes in the revised paper.

First, early in the introduction, we changed the sentence referring to echo-chambers to:

Problematic online encounters between both like-minded people and people with opposing views were shown to result in the reinforcement of extreme views and intensification of belief polarisation, reducing the willingness of groups to engage in constructive dialogue [3-

7]. In extreme cases, this can lead to radicalisation, where individuals or groups adopt extreme views that sanction political violence [8-10]. (see also above)

Second, on page 17-18, we now talk about a dual path to polarisation accounting for both intragroup and intergroup type of interaction. For example, we now added the following (referencing Bruns and Bail's work):

Thus, while people might prefer to interact online with people and media content reinforcing their view, social media platforms provide many opportunities to exposure to different and opposing views, much more so than would be possible in the real world [115]. However, intergroup interaction in these situations (exposure to outgroup's position) can also increase polarisation, as exposure to opposing views can lead to an increase one's own belief conviction [116] rather than encouraging reflection and change.

Comment 4: More engagement with the literature on affective polarisation.

"Second, a key issue with the paper is that it focuses on political polarization without engaging sufficiently with the literature in political science, the primary locus of debates around polarization and its drivers. Specifically, I believe this paper could be strengthened significantly by explicitly engaging with the ideas in affective polarization literature, and to better discuss the nature and foundation of polarization. This literature very explicitly builds on social psychology, and particularly social identity theory, and thus overlaps substantially with the aims of the paper at hand. The literature also has a significant focus on the relationship between identities and opinions. The paper currently makes various more or less explicit assumptions about the relationship between opinions and identities (e.g., opinions -> identities). I would suggest the author(s) to engage with the affective polarization literature, and see how and whether they would like to incorporate it, or perhaps nuance their claims and assumptions. The author(s) may find that the paper could become significantly stronger if it was theoretically situated within e.g., the affective polarization literature, allowing its contribution to be narrowed and sharpened. Here are some examples of recommended reading:

Iyengar, S., Lelkes, Y., Levendusky, M., Malhotra, N., & Westwood, S. J. (2019). The origins and consequences of affective polarization in the United States. *Annual review of political science*, 22, 129-146.

Mason, L. (2018). *Uncivil agreement: How politics became our identity*. University of Chicago Press.

Achen, C., & Bartels, L. (2017). *Democracy for realists: Why elections do not produce responsive government*. Princeton University Press.

Robison, J., & Moskowitz, R. L. (2019). The group basis of partisan affective polarization. *The Journal of Politics*, 81(3), 1075-1079.

Webster, S. W., & Abramowitz, A. I. (2017). The ideological foundations of affective polarization in the US electorate. *American Politics Research*, 45(4), 621-647.

Druckman, J. N., Klar, S., Krupnikov, Y., Levendusky, M., & Ryan, J. B. (2021). How affective polarization shapes Americans' political beliefs: A study of response to the COVID-19 pandemic. *Journal of Experimental Political Science*, 8(3), 223-234.

Our response: This is very helpful and in the revised manuscript, we are more explicitly discussing how the constructs of affective and issue-driven polarisation (based on the suggested literature, but also considering other relevant work) inform the conceptualisation of polarisation used in the paper. As such, in relation to how we conceptualise polarisation, we now say on page 3:

We adopt a conceptualisation of polarisation as a process of increasing the ideological and psychological distancing between (psychological) groups underpinned by specific social identity contents (21, 25-27). This conceptualisation integrates aspects of both issue-driven polarisation (where the distancing is driven by ideological dissent that can go across partisan boundaries) and affective polarisation (where the distancing driven by animosity and hostility between the opposing sides). As a result, ideological position becomes the driver of the process of sorting into opposing sides along the lines of conflicting multiple social identities, a process further enhanced by difference, distrust, and disdain for the opposing group [28, 29]. Taking this view, polarisation arises as a gradual extremization of the ingroup position and increased distancing from ideological outgroups.

Comment 5: Drawing more on previous work arguing for the use of ABMs in social psychology. “Third, the author(s) argue for the incorporation of group identities in agent-based models, and drawing on social psychology. This is a good point, but it has to some degree already been made, as there is an ABM literature in CSS focused on the role of identities in polarization, and the relationship between opinions and identities. This work argues explicitly for the need to draw on social psychology when modeling opinion dynamics, for instance in the work of Törnberg (whom you cite in other contexts):

Törnberg, P., Andersson, C., Lindgren, K., & Banisch, S. (2021). Modeling the emergence of affective polarization in the social media society. *PLoS One*, 16(10), e0258259.

Törnberg, P. (2022). How digital media drive affective polarization through partisan sorting. *Proceedings of the National Academy of Sciences*, 119(42), e2207159119.

Our response: We now more clearly integrate work by Törnberg and colleagues in our argument (including the recommended references). For example, we now say on page 3:

This conceptualisation integrates aspects of both issue-driven polarisation (where the distancing is driven by ideological dissent that can go across partisan boundaries) and affective polarisation (where the distancing driven by animosity and hostility between the opposing sides). As a result, ideological position becomes the driver of the process of sorting into opposing sides along the lines of conflicting multiple social identities, a process further enhanced by difference, distrust, and disdain for the opposing group [28, 29].

In the Discussion, we also refer to this work on page 21:

While some of these models effectively combine classic social identity and social influence theories and agent-based modelling [e.g., 28, 29], many fail to account for the critical impact of group dynamics and social identity on agent behaviours, as evident in recent contributions to the psychology of collective action and ideological polarisation [68, 130].

Comment 6: Details of the simulation in Figure 1. “Fourth, a more minor issue: Figure 1 is said to come from a simulation, but no details are given. I would suggest either including a full description of the simulation, or not including the figure”

Our response: We changed the figure, so it is now directly based on work from a previous modelling paper which includes in the text the details of the agent-based model that we used (please see: Betts, John M., and Ana-Maria Bliuc. "The effect of influencers on societal polarization." In *2022 Winter Simulation Conference (WSC)*, pp. 370-381. IEEE, 2022. <https://ieeexplore.ieee.org/abstract/document/10015491>):

Figure 1. Simulated evolution of an artificial society showing the stages of polarisation at the societal level. From left to right: the formation of camps, emergence of factions, and ultimately radical cells/extreme polarisation (adapted from [90]).

3rd Jul 24

Dear Ana-Maria,

Your Perspective titled "From Ideologically Opposed Camps to Radical Cells: Polarisation as the Gradual Fragmentation of a Divided Society" has now been seen by the same 2 referees as before. Their comments appear below. In the light of their advice I am delighted to say that we are happy, in principle, to publish it in Communications Psychology under a Creative Commons 'CC BY' open access license.

We will not send your revised paper for further review if, in the editors' judgement, the referees' comments on the present version have been addressed. If the revised paper is in Communications Psychology format, in accessible style and of appropriate length, we shall accept it for publication immediately.

EDITORIAL REQUESTS:

* Please review the requests in the attached Editorial Request Table and address each individual item.

* Please check whether your manuscript contains third-party images, such as figures from the literature, stock photos, clip art or commercial satellite and map data. If any of the display items in your manuscript (figures, tables, boxes or movies) include images that are the same as, or are adaptations of, previously published images, please fill in the Third Party Rights Table, and return to us when you submit your revised manuscript. This information will enable us to obtain the necessary rights to re-use such material. If we are unable to obtain the necessary rights to use or adapt any of the material that you wish to use, we will contact you to discuss alternative options.

* Communications Psychology uses a transparent peer review system. On author request, confidential information and data can be removed from the published reviewer reports and rebuttal letters prior to publication. If you are concerned about the release of confidential data, please let us know specifically what information you would like to have removed. Please note that we cannot incorporate redactions for any other reasons.

*If you have not done so already, please alert me to any related manuscripts from your group that are under consideration or in press at other journals, or are being written up for submission to other journals (see www.nature.com/authors/editorial_policies/duplicate.html for details).

** Figures

Please remove all figures from the main text and upload them individually, one figure per file. To ensure the swift processing of your paper please provide the highest quality, vector format, versions of your images (.ai, .eps, .psd) where available. Text and labelling should be in a separate layer to enable editing during the production process. If vector files are not available then please supply the figures in whichever format they were compiled in and not saved as flat .jpeg or .TIFF files. If your artwork contains any photographic images, please ensure these are at least 300 dpi.

* Competing interests

Please include a "Competing interests" statement after the References. Note that we ask authors to declare both financial and non-financial competing interests. For more details, see <https://www.nature.com/authors/policies/competing.html>. If you have no financial or non-financial competing interests, please state so: "The authors declare no competing interests."

SUBMISSION INFORMATION:

* Your paper will be accompanied by a two-sentence editor's summary, of between 250-300 characters, when it is published on our homepage. A draft summary is included in the Editorial Request Table.

In order to accept your paper, we require the following:

* A cover letter describing your response to our editorial requests.

* A separate document detailing your point-by-point response to any issues raised by our referees (please include the referees' comments in this document).

* The final version of your text as a Word or TeX/LaTeX file, with any tables prepared using the Table menu in Word or the table environment in TeX/LaTeX and using the 'track changes' feature in Word.

* Production-quality versions of all figures, supplied as separate files. Photographic images should be 300 dpi in RGB format (.jpg, TIFF or native Photoshop format) and any labels/scale bars included in a separate layer from the image. Line art, graphs and schemes should be vector format (.ai, .eps, .pdf); Adobe Illustrator files are preferred and will minimize production time. Any chemical structures or schemes contained within figures should additionally be supplied as separate Chemdraw (.cdx) files.

At acceptance, the corresponding author will be required to complete an Open Access Licence to Publish on behalf of all authors, declare that all required third party permissions have been obtained.

Please note that your paper cannot be sent for typesetting to our production team until we have received this information; therefore, please ensure that you have this ready when submitting the final version of your manuscript.

ORCID

Communications Psychology is committed to improving transparency in authorship. As part of our efforts in this direction, we are now requesting that all authors identified as 'corresponding author' create and link their Open Researcher and Contributor Identifier (ORCID) with their account on the Manuscript Tracking System (MTS) prior to acceptance. ORCID helps the scientific community achieve unambiguous attribution of all scholarly contributions. For more information please visit <http://www.springernature.com/orcid>

For all corresponding authors listed on the manuscript, please follow the instructions in the link below to link your ORCID to your account on our MTS before submitting the final version of the manuscript. If you do not yet have an ORCID you will be able to create one in minutes.

IMPORTANT: All authors identified as 'corresponding author' on the manuscript must follow these instructions. Non-corresponding authors do not have to link their ORCIDs but are encouraged to do so. Please note that it will not be possible to add/modify ORCIDs at proof. Thus, if they wish to have their ORCID added to the paper they must also follow the above procedure prior to acceptance.

To support ORCID's aims, we only allow a single ORCID identifier to be attached to one account. If you have any issues attaching an ORCID identifier to your MTS account, please contact the Platform Support Helpdesk.

[link redacted]

We hope to hear from you within two weeks; please let us know if the process may take longer.

Best wishes,

Marike

Marike Schiffer, PhD

Chief Editor

Communications Psychology

REVIEWERS' COMMENTS:

Reviewer #1 (Remarks to the Author):

I enjoyed reading this revised manuscript proposing a theory of societal polarization. This is a significant improvement from the original manuscript, and I applaud the authors' efforts. In my original review I made three central criticisms. The first was that terms were poorly defined, and in this regard I believe the manuscript has improved greatly. The second was that this framework doesn't provide explanations for why "polarization" does or does not happen, and in this regard I do not think the core issue has been addressed. However, in reading the revision (and rereading my original review) I no longer see this as an issue, so long as my third original concern can be addressed: that it's not clear what real-world social contexts this framework applies to.

To begin, I'm going to reference a way of thinking about theory from qualitative research, and I'll preface this by saying I wish this thought had come to me while writing my original review. In qualitative research, there's a distinction between "process" theories and "variance" theories. Variance theories are what dominate quantitative disciplines like social psychology, they are meant to explain variance in an outcome and help predict when something will or will not occur. Process theories, conversely, are about illuminating the sequences of causes and effect that lead to one single outcome. They are *not* about explaining variance, they're about providing novel insights about how something (one thing) developed. After reading the revision I don't see a way that this theory could serve as a variance theory. For example, the I found the addition on page 24 of a couple paragraphs about how this framework could provide testable hypotheses unconvincing. It mostly just restated the dynamics of the proposed model. So rather than push this paper toward doing something I don't think it's designed to do (explain the factors driving variance in polarization), I want to push the authors in another direction: toward being a paper that elucidates how dissent contributes to the process of high polarization/radicalization. And in order for this to be a strong "process theory" paper, it needs to be clearer what sequential, real-world phenomenon this theory applies to.

This is where my original third critic (restated above) comes in. The paper still feels like it's assuming this level of polarization is the norm (e.g., on pg 5 it states "Conceptually, this process continues until a limiting state of complete segregation between the polarized groups is reached). "Complete segregation" by no means described all, or even most, groups across societies. The string needed to tie this process theory together is a clearer articulation of the specific phenomenon being described, and I think I would best describe that as "extremely high levels of polarization." This paper seems interested primarily in that context alone: a context where "radicalization" is frequent enough to warrant an explanation.

This touches also on the one term I think is too loosely defined through the paper, "polarization" (and to be honest the extremely loose use of the term polarization in the literature more broadly is a pet-peeve of mine). Taking this process theory approach means that something I criticized in my original review, the seemingly contradictory way dissent is talked about at different levels of

analysis, is now a strength. It's a strength because that's the novel contribution of this process theory: the dynamic and countervailing ways in which dissent causes high levels of polarization in a society across levels of analysis. I'm imagining a sentence in the paper like "The purpose of this framework is to illuminate the unique role dissent plays, at different levels of analysis, in the process of radicalization in highly polarized societies." This makes it clear the paper is interested in a specific phenomenon, and its contribution is highlighting a how a process (dissent) leads to that phenomenon.

If the authors can do these things, (1) articulate the phenomenon being theorized about (high polarization/radicalization) and make it clearly demarcated (i.e., this paper is not about explaining why some societies are versus are not polarized), and (2) articulate how dissent is a critical processes in creating that phenomenon, and how dissent works differently at different levels of analysis and different stages of the phenomenon (camps vs factions vs cells), then the paper will be well situated to make a strong contribution. Much of #2 above is already in the paper, it's #1 that needs the most work.

One minor comment, the "Social process" column in Figure 3 seems just redundant with the level of analysis. I'd suggest being more specific.

Signed Review: Jeff Lees

Reviewer #2 (Remarks to the Author):

I wish to thank the authors for engaging so thoroughly with my comments, and for the substantial effort that they put into the revision. The paper is now much more nuanced throughout, and better supports its claims and assumptions with existing literature. I'm content with the revised version and support its publication.

Here are some minor notes from my reading:

I'm not sure that the Russian-Ukrainian conflict in Europe is the best example for what the authors want to illustrate. Most of what I read on the European debate on the war has emphasized how the "European" identity has been empowered by Russia's invasion of Ukraine (and how the European

identity was perhaps to some degree founded on the contrast to Russia.) On the opposing side stands parties and groups that are subject to influence campaigns from Russia. While there might be a polarization process taking place, these confounding factors might make it a poor example – as it also raises the challenging issue of the relationship between national identities/conflicts and political polarization, which you may want to avoid dealing with. If you do want to stick with this example, perhaps draw more on existing literature that more clearly evidence these dynamics.

- First line on page 5, “he” should be “the”.

- On page 10, "frame sanctioning" should perhaps be "frame for sanctioning"

- Page 13, skip the comma before [92].

Response to Reviewers

Reviewer #1

General Feedback: “I enjoyed reading this revised manuscript proposing a theory of societal polarization. This is a significant improvement from the original manuscript, and I applaud the authors’ efforts. In my original review I made three central criticisms. The first was that terms were poorly defined, and in this regard I believe the manuscript has improved greatly. The second was that this framework doesn’t provide explanations for why “polarization” does or does not happen, and in this regard I do not think the core issue has been addressed. However, in reading the revision (and rereading my original review) I no longer see this as an issue, so long as my third original concern can be address: that it’s not clear what real-world social contexts this framework applies to.”

Our response: We thank again the reviewer for the constructive feedback. We describe below how we addressed the remaining issue of the real-world social contexts to which the proposed framework applies. In particular, we explain how the framework applies in contexts where either extreme polarisation or radicalisation has already occurred or where there is a significant risk for these to occur. We have added the following text on page 14:

Our framework helps us understand (post-hoc) instances of extreme polarization and radicalization, including political violence. By examining how these processes developed in specific communities and socio-political conditions, we can design more effective preventive measures. Additionally, this framework can identify high-risk conditions for polarization and radicalization, enabling timely interventions before critical points are reached. While we do not maintain that dissent and polarization would always result in extreme outcomes, in divided societies, regardless of the end result, the process of polarization is likely to develop asymmetrically – as even from the initial stage when a divisive issue emerges, the resulting camps would differ in their alignment to the dominant political power. This is important, as it will likely determine how active and engaged camp members are. For instance, the collective goals of an ideological camp aligned to the dominant political power would be in line with the status quo in a society (e.g., climate sceptics in a society governed by political conservatives supportive of the oil industry). For the opposed ideological group, going against the status quo – thus requiring more effort to produce social change – the processes of fragmentation and further clustering are likely to be more intense, resulting in more diverse social identity content. Without having power on their side,

group members need to be more active and innovative in devising means to achieve group goals that challenge the status quo [97].

Comment 1: “To begin, I’m going to reference a way of thinking about theory from qualitative research, and I’ll preface this by saying I wish this thought had come to me while writing my original review. In qualitative research, there’s a distinction between “process” theories and “variance” theories. Variance theories are what dominate quantitative disciplines like social psychology, they are meant to explain variance in an outcome and help predict when something will or will not occur. Process theories, conversely, are about illuminating the sequences of causes and effect that lead to one single outcome. They are **not** about explaining variance, they’re about providing novel insights about how something (one thing) developed. After reading the revision I don’t see a way that this theory could serve as a variance theory. For example, the I found the addition on page 24 of a couple paragraphs about how this framework could provide testable hypotheses unconvincing. It mostly just restated the dynamics of the proposed model. So rather than push this paper toward doing something I don’t think it’s designed to do (explain the factors driving variance in polarization), I want to push the authors in another direction: toward being a paper that elucidates how dissent contributes to the process of high polarization/radicalization. And in order for this to be a strong “process theory” paper, it needs to be clearer what sequential, real-world phenomenon this theory applies to.”

Our Response: This is a very important observation. We have now more clearly articulate the purpose of the framework and its applicability. We have also deleted the repetitive sub-section on possible testable hypotheses on page 24.

We further say on pages 23-24:

A practical implication of this work is that understanding polarization as a dynamic process driven by human interaction gives support to the use of social interventions to slow, interrupt, or remediate extreme polarization. For example, the use of social influence to attract people toward a more central view on issues [90], moderating communications within and between groups or introducing superordinate goals relevant to both sides of a conflicting issue are potential candidates for preventive programs or depolarizing interventions. (...)By modelling the transformation process we propose using real data, it may be possible to identify communities that might be at high

risk of extreme polarization or about to reach a tipping point in their evolution (for example, from moderate camp to extreme faction). Thus, on one level, this framework can be further developed to identify the conditions for political extremism to develop, on another level, it can be applied to assess the effectiveness of current strategies to address extremism and political violence, or design new ones.

Comment 2: “This is where my original third critic (restated above) comes in. The paper still feels like it’s assuming this level of polarization is the norm (e.g., on pg 5 it states “Conceptually, this process continues until a limiting state of complete segregation between the polarized groups is reached). “Complete segregation” by no means described all, or even most, groups across societies. The string needed to tie this process theory together is a clearer articulation of the specific phenomenon being described, and I think I would best describe that as “extremely high levels of polarization.” This paper seems interested primarily in that context alone: a context where “radicalization” is frequent enough to warrant an explanation.”

Our Response: This is an accurate assessment of one of the main arguments in our paper: the proposed framework is meant to explain the processes leading to “extremely high levels of polarization” and radicalization. We are not suggesting that once the process of polarisation starts, the outcome will invariably be extreme polarization, but when we have cases of extreme polarization in society, we can trace back the process and its roots by applying this framework. Further, as mentioned above, this framework can be used to detect conditions and risks and ultimately prevent outcomes such as extreme polarization and radicalization. We have now qualified the statement on pages 4- 5 to read as follows:

While in reality, extreme polarization is relatively rare, conceptually, this process could continue until a limiting state of complete segregation between the polarized groups is achieved, and radical cells may form. To understand how to slow down or stop this process, we need to approach polarization as a complex and progressive process driven by many interconnected factors, transforming both the society and individuals.

Comment 3: “This touches also on the one term I think is too loosely defined through the paper, “polarization” (and to be honest the extremely loose use of the term

polarization in the literature more broadly is a pet-peeve of mine). Taking this process theory approach means that something I criticized in my original review, the seemingly contradictory way dissent is talked about at different levels of analysis, is now a strength. It's a strength because that's the novel contribution of this process theory: the dynamic and countervailing ways in which dissent causes high levels of polarization in a society across levels of analysis. I'm imagining a sentence in the paper like "The purpose of this framework is to illuminate the unique role dissent plays, at different levels of analysis, in the process of radicalization in highly polarized societies." This makes it clear the paper is interested in a specific phenomenon, and its contribution is highlighting a how a process (dissent) leads to that phenomenon. If the authors can do these things, (1) articulate the phenomenon being theorized about (high polarization/radicalization) and make it clearly demarcated (i.e., this paper is not about explaining why some societies are versus are not polarized), and (2) articulate how dissent is a critical processes in creating that phenomenon, and how dissent works differently at different levels of analysis and different stages of the phenomenon (camps vs factions vs cells), then the paper will be well situated to make a strong contribution. Much of #2 above is already in the paper, it's #1 that needs the most work."

Our Response: We have made changes throughout the paper (please see also our responses to the comments above) to address point #1. In line with the reviewer's specific suggestion, we also added on page 20:

The purpose of this framework is to illuminate the unique role dissent plays in the process of polarization at different levels of society – a process that can culminate in extreme polarization and radicalization. The framework can be applied to contexts where extreme polarization has already occurred, is emerging, or where there is a high risk of it happening in the future.

Comment 4: "One minor comment, the "Social process" column in Figure 3 seems just redundant with the level of analysis. I'd suggest being more specific."

Our Response: We have amended the label of that column from "Social process" to "Key Change Mechanism".

Reviewer #2

General Feedback: “I wish to thank the authors for engaging so thoroughly with my comments, and for the substantial effort that they put into the revision. The paper is now much more nuanced throughout, and better supports its claims and assumptions with existing literature. I’m content with the revised version and support its publication. Here are some minor notes from my reading.”

Our Response: We thank the reviewer for the positive assessment of our revised paper and the constructive feedback.

Comment 1: “I’m not sure that the Russian-Ukrainian conflict in Europe is the best example for what the authors want to illustrate. Most of what I read on the European debate on the war has emphasized how the “European” identity has been empowered by Russia’s invasion of Ukraine (and how the European identity was perhaps to some degree founded on the contrast to Russia.) On the opposing side stands parties and groups that are subject to influence campaigns from Russia. While there might be a polarization process taking place, these confounding factors might make it a poor example – as it also raises the challenging issue of the relationship between national identities/conflicts and political polarization, which you may want to avoid dealing with. If you do want to stick with this example, perhaps draw more on existing literature that more clearly evidence these dynamics.”

Our Response: We agree with the reviewer that a better example to illustrate our point can be used. We have removed the example about the Russian-Ukrainian conflict and included instead an example on dissent about global warming. We now say on page 13:

The role of dissent in fracturing societies is clearly shown in the global warming debate, which has polarized the public into opposing ideological camps. Although, broadly speaking, there is scientific consensus on the anthropogenic causes of global warming, the public is divided into those self-identifying as climate change sceptics, in line with a narrative that contradicts most scientific evidence, and an ideological camp of people who support a narrative aligned to climate change science. Focusing on the supporters of mitigating climate change, there is fragmentation within emerging

factions (some more extreme than others) which not only endorse pro-environmental behaviors and political action, but also engage in radical and sometimes violent action, labelled as eco-terrorism.

Comment 2/Editing comments:

- First line on page 5, “he” should be “the”.
- On page 10, "frame sanctioning" should perhaps be "frame for sanctioning"
- Page 13, skip the comma before [92].

Our Response: We have addressed all the editing comments and conducted one more detailed editing of the manuscript.